# Determination of Conformational and Functional Stability of Potential Plague Vaccine Candidate in Formulation

**DOI:** 10.3390/vaccines11010027

**Published:** 2022-12-23

**Authors:** Krubha Athirathinam, Selvasudha Nandakumar, Shailendra Kumar Verma, Ruckmani Kandasamy

**Affiliations:** 1Department of Pharmaceutical Technology, Centre for Excellence in Nano-Bio Translational Research (CENTRE), University College of Engineering, Bharathidasan Institute of Technology, Anna University, Tiruchirappalli 620024, India; 2Department of Biotechnology, Pondicherry University, Puducherry Union Territory, Puducherry 605014, India; 3Microbiology Division, Defense Research and Development Establishment (DRDE), Gwalior 474002, India

**Keywords:** protein formulation, vaccine, drug–excipient interaction, stabilization, polymeric biomaterial

## Abstract

Generally, protein-based vaccines are available in liquid form and are highly susceptible to instability under elevated temperature changes including freezing conditions. There is a need to create a convenient formulation of protein/peptides that can be stored at ambient conditions without loss of activity or production of adverse effects. The efficiency of naturally occurring biocompatible polymer dextran in improving the shelf-life and biological activity of a highly thermally unstable plague vaccine candidate protein called Low Calcium Response V antigen (LcrV), which can be stored at room temperature (30 ± 2 °C), has been evaluated. To determine the preferential interactions with molecular-level insight into solvent–protein interactions, analytical techniques such asspectroscopy, particle size distribution, gel electrophoresis, microscopy, and thermal analysis have been performed along with the evaluation of humoral immune response, invivo. The analytical methods demonstrate the structural stability of the LcrV protein by expressing its interaction with the excipients in the formulation. The invivo studies elicited the biological activity of the formulated antigen with a significantly higher humoral immune response (*p*-value = 0.047) when compared to the native, adjuvanted antigen. We propose dextran as a potential biopolymer with its co-excipient sodium chloride (NaCl) to provide protein compactness, i.e., prevent protein unfolding by molecular crowding or masking mechanism using preferential hydrophobic interaction for up to three weeks at room temperature (30 ± 2 °C).

## 1. Introduction

Protein drugs and therapeutics such as vaccines are increasing steadily and grabbing attention among drug developers due to their selectivity, specificity, and biocompatibility. The activity of protein pharmaceuticals depends on its structure, and hence any alterations in the structure or conformational change are responsible for the difference in the three-dimensional (3D) structure of the protein, which can lead to unexpected results including protein instability and other adverse effects such asallergic reactions, etc. Proteins consist of multiple polypeptide chains termed protein subunits, which may be monomeric, dimeric, or multimeric. These complex protein structures are stabilized by various interactions such ashydrogen bonds, disulfide bonds, hydrophobic interactions, electrostatic interactions, and van der Waals forces [1]. Although these biological interactions are essential to managing the 3D structure of the proteins, therapeutic/prophylactic proteins such asvaccines, when exposed to different environmental changes including temperature, pH, mechanical agitation, etc., are prone to lose their activity; thus, the storage and transportation become challenging. This can be harmful or even fatal for patients who take the medications and can also increase costs because of the requirement of a cold chain [2].

In this present study, we have characterized the recombinant LcrV protein of *Yersinia pestis* for its stability. Earlier, LcrV has been characterized as a potential vaccine candidate in formulation with the capsular antigen (F1) against the plague in a mouse model [3]. Plague—a zoonotic infection—remains to be one of the fatal diseases caused by a Gram-negative bacterium called *Yersinia pestis*. The plague has been considered as a re-emerging infectious disease by the World Health Organization (WHO) with possibilities of being used as a bioweapon [4], like the pneumonic form of the plague transmits from human to human [5]. The Centre for Disease Control (CDC) [6] and Stanford University report [7] have categorized *Y. pestis* as a biosafety level-3 pathogen, and around 5000 human plague cases are reported every year globally [8]. Essential antibiotics and supportive care are the available medical aids to cure the plague if diagnosed at the earliest, as no commercial vaccines are available against the plague to date. Sub-unit recombinant vaccines are under trial, targeting the two virulence protein factors F1 and LcrV of *Y. pestis* to boost immunity against the infection [9]. These proteins suffer rapid instability when exposed to a temperature above 4–8 °C, i.e., they deactivate at stressful conditions, leading to loss of potency of the vaccine, and hence remain challenging. The general reason proposed for the instability of the proteins is the transition from a folded to unfolded state in unfavorable conditions. The unfolding of proteins (and of protein vaccines) leads to alterations in the protein structure and subsequent aggregation of partially denatured proteins, causing unfavorable thermodynamic interactions [10]. Chemical instability owing to unwanted reactions such as hydrolysis, oxidation, deamination, and breakage or formation of disulfide bonds can also result in loss of vaccine potency. All these destabilizing processes are influenced by factors such as pH, buffer, salts, etc., and are accelerated by fluctuations in temperature.

To overcome these shortcomings, various stabilization methods viz. low-temperature storage, freeze-drying, and the addition of various excipients such asosmolytes, additional amino acids, salts, sugars, and polymers are used [11]. A proper understanding of the interactions between proteins and these excipients is essential to designing an optimal formulation. Protein drug formulations are in combat due to the lack of a good understanding of the protein’s structure and its conformational characteristics. Though the four classes of protein structures (primary, secondary, tertiary, and quaternary structure) are widely studied, protein structure prediction in formulations remains challenging. Among different stabilization methods, polymeric materials that are capable of stabilizing biomolecules at room temperature and to agitation are of significant interest. Various kinds of polymers have been shown to stabilize proteins. Stabilization is generally due to one or more properties of polymers, such assurface activity, steric hindrance of protein-protein interaction, preferential exclusion, and increased viscosity that limit the protein’s structural movement [12]. The use of polymers as stabilizers appears to be one of the most efficient methods because of their solubilizing property and safety for parenteral administration.

We aim to briefly cover the behavior of our target protein LcrV in the formulation, mainly the protein–polymer interaction using feasible analytical tools to analyze the structural stability of the protein in the formulation. This work also investigates the biological activity of the formulated protein stored at room temperature (30 ± 2 °C) for a reasonable period. The mechanisms involved in stabilization by the naturally occurring biocompatible polymer dextran and sodium chloride in improving the stability and shelf life of LcrV antigen, a promising vaccine candidate against the plague, have also been investigated.

## 2. Materials and Methods

### 2.1. Ethics Statement

All procedures performed in studies involving mice were under the ethical standards of the institution at which the studies were conducted and followed the recommendations in the ARRIVE guidelines following the U.K. Animals (Scientific Procedures) Act, 1986 and associated guidelines, EU Directive 2010/63/EU for animal experiments. The Institutional Animal Ethics Committee (IAEC) authorization was obtained from Alagappa University, Karaikudi, Tamil Nadu, India, and the approval number is IAEC/AU/2019/7. All the guidelines and regulations of good laboratory animal care were followed during the experiment. The experimental mice were maintained according to the recommendations of the committee for control and supervision of experiments on animals (CPCSEA), Government of India.

### 2.2. Materials

The purified LcrV antigen was kindly received from the Division of Microbiology, DRDE, Gwalior which was dispersed in recommended native protein buffer (pH = 7) containing 50 mM of sodium dihydrogen phosphate (NaH_2_PO_4_) and 50 mM of sodium chloride (NaCl) and sterilized by passing through a 0.22 μm syringe filter. The protein concentration was 1.5 mg/mL and was determined using the Bradford assay and Bovine Serum Albumin was used as a reference. All reagents and excipients were purchased from Sigma Aldrich. All excipients were of high purity grade (>99%). The entire procedure was performed using Milli Q water.

### 2.3. Methods

#### 2.3.1. LcrV Formulations

The optimized formulation of LcrV (PD1) was prepared using Dextran (70,000 Dalton) and NaCl as stabilizing agents after various trials. The protein and polymer were taken in a 1:1 *w/v* ratio for the final formulation (with a protein concentration of 150µg/mL). Mild agitation (150 rpm) at 4–8 °C for four hours was provided to the sample to enhance the reactivity of LcrV with co-excipient (150 mM NaCl) first. Then, the polymer (Dextran 70) was added to the above solution and allowed for reactivity at 150 rpm for four hours at 4–8 °C. Immediately after four hours, the samples were used for experiments after incubation at room temperature (RT) i.e., 30 ± 2 °C, and were directly used for experiments, periodically after estimating the protein content using Bradford Assay. The LcrV in the formulation stored at room temperature (30 ± 2 °C) for three weeks was compared with the native LcrV stored at the optimal environment (4–8 °C) for its stability and biological activity.

#### 2.3.2. Rheology Measurements

The rheology of dextran and formulations was performed at 25 °C to assess the injectable and syringeable properties. The rheological behavior was studied with the help of a Modular Compact Rheometer (MCR 302, Anton Paar, Ashland, VA, USA) equipped with a cone-plate sensor CP 75-1 (with a diameter of 74.982 mm, a cone angle of 0.998, and truncation of 148µm). The plate temperature (25.0 ± 0.1 °C) was controlled by the truly Peltier temperature-controlled system. The samples were incubated at 25.0 ± 0.1 °C for 5 min in a thermostated bath, and 2 mL (1.95 mL) of samples were immediately placed on the plate for rheological measurements. In each experimental run, 25 measuring points were measured with a 10 s interval [13].

#### 2.3.3. UV–Visible and Fluorescence Spectroscopy

The absorption spectra were taken using a UV-2700 spectrophotometer (Shimadzu, Kyoto, Japan) equipped with 1.0 cm quartz cells. The samples were transferred to a UV transparent quartz cuvette and scanned at a wavelength range of 200–400 nm at 25 °C [14]. Fluorescence spectroscopy of the formulations was carried out on a JASCO fluorescence spectrometer attachment with the Peltier-type temperature control system. Intrinsic fluorescence emission spectra were collected at 30 °C in a 1 cm path length quartz cuvette. The excitation wavelength was 280 nm with a 1 nm bandwidth. The emission spectra were collected from 300 to 400 nm with a bandwidth of 10 nm. The intensity of fluorescence was corrected by subtracting the blank (without protein) [15].

#### 2.3.4. Fourier Transform Infrared Spectroscopy (FTIR)

FTIR was analyzed for all the samples using the attenuated total reflection (ATR) method in Jasco FT/IR 6300 at a frequency range of 4000–400 cm^−1^. The second derivative spectra were also measured for protein structure analysis [16].

#### 2.3.5. Particle Size Distribution and Zeta Analysis

The particle size and zeta potential of the samples were assessed using Malvern Zetasizer -Nano ZS (Malvern Instruments Ltd., Worcestershire, UK). The dynamic light scattering (DLS) technique was used to determine the particle size and the electrophoretic light scattering (ELS) for zeta potential. A disposable glass cuvette and a capillary cell were used to hold the samples for size and zeta respectively, and the experiment was performed in triplicates [17].

#### 2.3.6. Polyacrylamide Gel Electrophoresis (PAGE)

Sodium Dodecyl Sulfate (SDS)-PAGE was performed for pure LcrV and LcrV formulation PD1 (stored at RT) using 12% *w/v*, acrylamide separating gel, and a 4% *w/v*, stacking gel both containing 0.1% *w/v*, SDS [18]. Non-denaturing polyacrylamide gel electrophoresis was also performed [19]. Following Coomassie brilliant blue staining, the gels were photographed with Bio-Rad Gel Doc EZ imager (Bio-Rad Laboratories, Hercules, CA, USA), and the photographs were analyzed and annotated using Image lab 4.1 software (Bio-Rad Laboratories, Hercules, CA, USA) to determine each protein band.

#### 2.3.7. Circular Dichroism

Analysis of thermal denaturation was performedon a JASCO J-815 CD spectrometer (Japan) for protein and formulation samples in a 1 mm quartz cuvette. Circular dichroism spectra were collected from 300 to 200 nm. A scanning speed of 50 nm/min was used with a 1-s averaging time and a 0.1 nm step at a temperature of 25 °C. The signal was averaged over 10 scans corrected by subtraction of the spectra acquired on buffer alone [20].

#### 2.3.8. Transmission Electron Microscopy

Selected samples were applied to copper grids by placing a drop of the sample in a parafilm-lined petri dish along with at least 300 μL of mother liquor from the sample’s origin close to the sample drop. After the required steps, TEM images were acquired using an FEI Tecnai T12 electron microscope with an FEI Eagle 4k × 4k CCD camera and operating at 120 kV using a single-tilt specimen holder. Images were collected at nominal magnifications of 6500× (6.6 nm/pixel), 21,000× (0.5 nm/pixel), and 52,000× (0.21 nm/pixel) using the automated image acquisition software package Leginon (FEI, Hillsboro, OR, USA) [21].

#### 2.3.9. Differential Scanning Calorimetry

Differential Scanning Calorimetry was performed using a protein concentration of 1.5 mg/mL with a DSC—8000 differential scanning calorimeter (Perkin Elmer, Waltham, MA, USA). The samples were crimped in a standard aluminum pan and DSC thermograms were obtained from 0 °C to 150 °C at a scan rate of 10 °C/min under constant purging of nitrogen at 20 mL/min [22].

#### 2.3.10. Immunization of Mice

To assess the biological activity (mainly the protective efficiency and immune response) of PD1—the formulated vaccine candidate stored at room temperature (30 ± 2 °C) for three weeks, female Balb/C mice (6 weeks old) were collected and were divided into four groups (5 mice/group), i.e., Control group; Blank group (containing the excipients alone); Adjuvanted LcrV group (pure LcrV with aluminum hydroxide gel at 0.35% in sterile phosphate buffer saline) [3]; and Formulated LcrV (PD1) group (LcrV stabilized at 30 ± 2 °C for three weeks).

The animals were utilized for the evaluation of humoral immunity (IgG antibody) against the LcrV antigen of *Y. pestis*. Animals were immunized subcutaneously with 20 μg/mouse of the protein [3]. The animals of the control group received phosphate-buffered saline (PBS) only, and the animals of the blank group received the excipients alone. The immunizations were given on the 0th day, 14th day, and 21st day. All the immunized animals were subjected to blood collection on days 0, 21, and 28, and sera were separated to compare the (humoral) immune response of pure LcrV (adjuvanted) stored at 4–8 °C with formulated LcrV stabilized at 30 ± 2 °C for three weeks.

#### 2.3.11. IgG Antibody Immune Response

The endpoint titers of IgG antibodies in the serum of vaccinated and control group animals were evaluated using an enzyme-linked immunosorbent assay (ELISA) by following the previously reported protocol [3]. Briefly, LcrV protein (100 ng/well) was coated in 96-well sterile individual ELISA plates in 0.05 M carbonate buffer, pH 9.6. Overnight at 4 °C, the coated plates were incubated. The next day, after three washings with 0.05% Tween 20 in PBS (PBS-T), the plates were blocked with 5% skimmed milk powder in PBS and incubated for 2 h at 37 °C. Again, following extensive PBST washings (five times), the first and second booster test sera from immunized and control groups were serially diluted and added in triplicate wells (100 μL/well) of the respective plates. The plates were incubated for 1 h at 37 °C. Plates were then probed with anti-mouse horseradish peroxidase (HRP) labeled IgG (Sigma, Burlington, MA, USA) raised in the rabbit at 1:20,000 dilutions in PBS and incubated for 1 h at 37 °C. The plates were washed as earlier and the reaction was developed with3, 3′, 5, 5′-Tetramethylbenzidine (TMB) as substrate and after 10 min stopped by 2N Sulfuric acid. The optical density (OD) was measured at 470 nm by a multimode plate reader (EnSpire—Perkin Elmer, Waltham, MA, USA).

#### 2.3.12. Statistical Analysis

The statistical data were analyzed by the Mann–Whitney U and Kruskal–Wallis non-parametric tests. In the results, the differences were considered to be statistically significant when the *p*-value was lower than 0.05. All statistical analysis was conducted using GraphPad Prism (San Diego, CA, USA) 6.0 version.

## 3. Results and Discussion

### 3.1. Rheology

As the rheological characteristics of biopolymer solutions are complex and are affected by the experiment conditions, the biopolymer dextran alone and the protein formulation (PD1) were evaluated for their flow property using rheology measurement. Both the elastic and viscous modulus tended to decrease successively as the shear rate increased to 4 (Figure 1a) in dextran. Beyond this shear rate, Newtonian curvature was observed. In the case of the formulation also (Figure 1b), Newtonian curvature was observed with an increasing shear rate (12–100 pa). Dextran and the formulations expressed liquid-like behavior at 30 °C. Hence, the formulation and the polymer are compatible with each other, and polymer viscosity does not influence the syringeable and injectable properties of the vaccine formulation. Thus, this data attributes to the additive property of dextran to LcrV.

### 3.2. UV-Visible Spectroscopy

A hypochromic effect on the intensity maxima of protein formulation was observed at 280 nm within 0–1 h due to the complex formation of LcrV with the polymer, when compared to native LcrV protein. This result indicates an interaction between the antigen and the excipients to offer stability, though they are not conjugated chemically [23]. Simply, the addition of excipients decreased the absorption intensity, but not the absorption maxima, expressing the interaction between LcrV and dextran by the binding leading to n–π* transition of Tyrosine (Tyr) and Tryptophan (Trp) fluorophores (an important aspect in detecting protein stability) [24,25,26]. Hence, no bathochromic or hypsochromic shifts in the protein formulations are observed, ensuring the conformational state of the protein in the formulation kept at 30 ± 2 °C for three weeks. The interaction could be between the amino group of LcrV and the OH group of dextran, i.e., the tyrosyl side chain of LcrV contributes phenolic hydroxyl groups to the stability of the tertiary structure of protein via intramolecular hydrogen bonding and hydrophobic interaction with the local environment. Hence, the aromatic amino acid peaks can be observed in the formulation at 280 nm, as shown in Figure 2, which reveals the stability of the protein.

### 3.3. Fluorescence Spectroscopy

For the formulation at room temperature (30 ± 2 °C), the n–π*transitions of fluorophores groups (Trp or Tyr) were measured and compared using fluorescence spectroscopy. The emission maxima wavelengths (λmax) and the ratio of the intensity at 345 nm (Figure 3a) give an indication of the compactness of the LcrV in the formulation expressing the exposure of Trp^61^ and Trp^186^ of LcrV in the solvent. The fluorescence intensity decreased in the formulation after the addition of stabilizers without any red/blue shift in the emission maxima indicating a stable microenvironment around the amino acid residues [14,15,26] even after exposure to 30 ± 2 °C for three weeks. The reduction in the fluorescent intensity can be attributed to the molecular interaction of LcrV or quenching, leading to the shielding of the amino acid group of LcrV by polymer to offer stability, which has been further evaluated using the FTIR method discussed below. Thus, this shielding may provide a more compact and stable native structure to LcrV [15,25]. The emission wavelength obtained in the formulation is in agreement with the excitation wavelength (Figure 3b) and UV absorption spectra of pure LcrV at 280 nm.

### 3.4. Fourier Transform Infrared Spectroscopy (FTIR)

The molecular interaction of LcrV with the polymer has been analyzed using FTIR. From the data (Figure 4), the peak at around 3400 cm^−1^ corresponds to the -OH stretching of dextran and -NH stretching of the amino group of LcrV. As the peak at 3459 cm^−1^ has broadened, it could mean that a favorable interaction has occurred, usually an intermolecular hydrogen bonding (-OH stretching of the carboxylic group) between LcrV and dextran, i.e., H-O-H bond. The reduction in the peak at 2342 cm^−1^ in formulations corresponds to the interaction between the amino group of LcrV (i.e.,) -C≡N and -C≡C stretching with the -OH group of the dextran to offer stability, which is in agreement with the spectroscopic results confirming the n–π* transition in the formulation.

The common peak found in formulations, pure protein, and the polymer at 1639 cm^−1^ represents the Amide I components region corresponding to secondary structure conformation of proteins and C=O stretching in the case of dextran. This is the most widely analyzed region of the protein to ensure conformational stability. Hydrogen bonding of the carbonyl group to the amide group in the peptide yields information on the secondary structures of a protein. Hence, the second derivative Amide I (C=O stretching) regions were evaluated for the presence of alpha-helix, coils, and beta sheets (Appendix A) in the third week. The formulation PD1, even after exposure to 30 ± 2 °C for three weeks, expressed the corresponding peaks (Appendix A), proving the conformational state, i.e., stable state of LcrV in the formulation. Amide II (C-N stretching) was also evaluated (Appendix A), from which, apart from pure LcrV, both the formulation and pure polymer did not show any peak at 1600–1500 cm^−1^, ensuring the interaction of the protein with the stabilizers to offer stability.

### 3.5. Particle Size Distribution Analysis

To evaluate any aggregation of LcrV in formulation, particle size distribution was performed. From the data (Figure 5), it is clear that the formulation (PD1) has shown 100% size distribution (by intensity) after an hour to three weeks (Figure 5d–f) i.e., the actual particle size of LcrV [27] (Figure 5a) in formulation increased to 588 ± 2.71 d.nm represented by a single peak in size distribution curve [28] (Figure 5d) proving the molecular crowding of the protein by the polymer and the co-excipient to provide stability. Though an increase in particle size could mean an aggregation, in our case it is due to the interaction of LcrV with dextran [26]. That is, the polydispersity index (PDI) value of the formulation between 0 and 1 h (Figure 5c) was around 0.67 ± 2.83 and converted to a mono (homogenous) dispersion with a PDI value of 0.032 ± 1.59, an acceptable range for polymer-based drug delivery applications [28,29], at the end of three weeks. Thus, the result emphasizes the stability of LcrV in the formulation and not aggregation. In general, aggregation occurs when protein particles tend to come together strongly since a large surface area of particles results in high surface energy [30]. The protein of interest (LcrV) carries electrostatic charges that are dependent on the surrounding solvent phase. After the fabrication of LcrV formulations at pH 7, all particles existed in the ordered form by the excluded volume effect and molecular masking of dextran, preventing the particles from coming too closely into contact with each other; thus, aggregation hardly occurs.

An attempt was made to check the stability of LcrV formulation with NaCl alone, where LcrV particles became unstable during storage at the end of 8 days due to the weakening of the ionic bond between salt and protein. Heterogenic distribution (aggregation) of LcrV was observed by exerting multiple peaks [31] (Appendix A). Though the protein remains stable at RT for eight days, this aggregation is prone to instability. Apparently, in the case of formulation with dextran alone (data not shown), the zeta potential of both protein and the polymer is negative (Appendix A). The interaction of a protein with polymers and other biomaterials has characteristic properties, such as the local electrostatic charge distribution and the electrical double layer potential, which play a significant role in defining the biological interactions, aggregation behavior, and stability [32]. Hence, the negative zeta potential values of both protein and polymer initiate electrostatic repulsion [33,34], i.e., like charges repel each other, thus preventing protein aggregation. However, this repulsive force become sunstable soon due to weak interaction between them or bond dissociation. Hence, LcrV with dextran alone is also expected to have temporary stability only. Thus, protein stabilization occurs only with the combination of the stabilizer (dextran) with its co-stabilizer (NaCl), for a longer period. In our case, the zeta potential value of the final PD1 formulation (−23 mV) typically expresses a high degree of stability (Appendix A), emphasizing the stability of LcrV in solution and not aggregation [34].

### 3.6. Gel Electrophoresis

SDS-PAGE was performed to ensure the structural stability of formulated LcrV at room temperature (30 ± 2 °C). Degradation of unformulated LcrV (Figure 6a) was observed after exposure to 30 °C for 24 h. The formulation (PD1) showed bands corresponding to LcrV (37 kDa) antigen in SDS PAGE when performed after different time intervals (from 1 h to 3 weeks) with similar band patterns (Figure 6b–e). After three weeks of exposure to room temperature (30 ± 2 °C), degradation was observed in the formulation (marked as PD1 O in Figure 6f) represented by a reduction in molecular weight. Thus, this result again manifests the stability of the protein in the formulation for three weeks and proves the stabilization potential of the polymer dextran along with NaCl.

Similarly, SDS-PAGE was also performed to evaluate the stabilization efficacy of NaCl alone towards LcrV. This showed that NaCl was able to provide stability to LcrV at 30 ± 2 °C for only eight days. The banding pattern of the NaCl-based formulation as highlighted in Figure 6g ensures the reduction in the molecular weight of the protein (LcrV), thus confirming the degradation of the protein of interest after eight days, in agreement with particle size distribution data. This again proves that dextran and NaCl in combination remain to be efficient for the stabilization of LcrV at room temperature (30 ± 2 °C) for a longer period. Native PAGE, which omits the denaturing conditions, was also performed to ensure the stability of the LcrV protein (37 kDa) in the formulation. Similar band patterns were observed in Native PAGE also (data not shown).

### 3.7. Circular Dichroism

Secondary structure was determined by CD spectroscopy in the “far-UV” spectral region (190–250 nm). These wavelengths are attributed to the peptide bond, and the chromophore signal arises only when the protein is located in its regular, folded setting. Rarely, the unfolded form of some protein will acquire a clear but different secondary structure than its native form. The spectra of PD1 in the far-UV region when compared to that of the native LcrV display the amide chromophores, representing the stability of the secondary structure of LcrV even after exposure to room temperature (30 ± 2 °C) for three weeks. Here, Figure 7 shows the characteristic spectra of α-helix with minima at 208 and 222 nm (attributed to hydrogen bonding environment) and a positive band at ~190 nm; β-sheets with a negative band between 210–220 nm and positive band at around 198 nm; and random coils with a negative band around 200 nm [26]. The formulation PD1 (Figure 7b) has shown the secondary structure compactness as in pure LcrV (Figure 7a). There were no drastic shifts in the percentage of α-helix between native LcrV (24% α-helix) and formulated LcrV after three weeks of storage at 30 ± 2 °C (20% α-helix), ensuring the stability of the protein in the formulation.

### 3.8. Transmission Electron Microscopy

The TEM analysis of pure LcrV protein and PD1 formulation was performed to ensure the particle size of protein in the formulation, as masking of the protein by the stabilizers was observed in all the analyses. The TEM confirms the presence of protein (LcrV) in the formulation. From Appendix A, it is clear that protein (LcrV) is expressed as dark spots with small particle sizes.

### 3.9. Differential Scanning Calorimetry

DSC is a micro-calorimetric method used to differentiate a biomolecule’s stability in its native form. The measurement of the thermal transition midpoint (Tm) indicates the stability of that particular biomolecule. When the Tm is higher, the biomolecule will be more stable, thus the Tm is a reflection of the protein’s thermal stability [35]. The DSC thermogram (Figure 8) depicts the Tm change of LcrV in PD1 formulation when compared to native LcrV (stored at 4 °C) upon increased temperature storage (at 30 °C for three weeks). The result shows improved stability of the folded form of LcrV at higher temperature storage for three weeks. The transition with the Tm at the protein’s melting temperature, i.e., 48 °C was chosen to evaluate the efficacy of the excipients (dextran and NaCl) in enhancing the stability. A positive transition (from 48 °C to 63 °C) in the stability of LcrV was observed in the formulation, expressing the direct interaction between the protein and the excipients to offer stability. Additionally, the amount of heat required to convert LcrV from its native state to the unfolded state (enthalpy of unfolding (ΔH)) increased when the dextran and NaCl were added, reflecting the degree of the overall intramolecular interactions [36]. This result signifies the preferential binding between the protein and the excipients to improve the compactness of LcrV in the formulation, thus improving protein thermostability.

### 3.10. IgG Antibody Response

To measure the IgG endpoint titers, sera were collected 7 days after the first and second injections, respectively, from the immunized animals (five animals per group). The cut-off value for the assays was calculated as the mean OD (+2 s.d.) from sera of the control group assayed at 1:200 dilutions. The IgG endpoint titers were calculated as reciprocal of the highest serum dilution giving an OD more than the cut-off. The IgG endpoint titer in the sera of both pure LcrV (adjuvanted) and PD1 formulation (injected after storage at 30 ± 2 °C for three weeks) was observed at 1.28 × 10^5^ and 2.56 × 10^5^ after the first and second booster, respectively (Figure 9). On the other hand, the animals injected with degraded LcrV (LcrV kept at 30 ± 2 °C for 24 h) did not show any immune response (data not shown). The immune response of PD1 immunized sera significantly improved when compared to the (adjuvanted) LcrV immunized sera after the first booster (* *p*-value = 0.047) in the endpoint titer. After the second booster, though there is nota significant difference between the immune response of PD1 and the native adjuvanted LcrV, the IgG endpoint titer was found to be similar (2.56 × 10^5^) in both cases. This result proves the biological activity of the stabilized LcrV after storage at room temperature (30 ± 2 °C) for three weeks. It is also noted that the formulation was able to provide a promising humoral immune response without the addition of an adjuvant, in comparison to the alum adjuvanted LcrV antigen stored at 4–8 °C. This highlights the fact that our proposed biopolymer dextran (70,000 Da) is not only capable of acting as a stabilizer but also has some adjuvant properties.

### 3.11. Mechanism of Protein Stabilization

An approach to reduce reliance on the proper implementation of the cold chain to protect plague vaccine candidate LcrV from degradation and inactivation at room temperature (30 ± 2 °C) has been evaluated. According to the literature survey, LcrV has two independently folding units with an N-terminal and a more stable C-terminal domain associated with coiled-coils [37,38]. In other work, thermal studies, including DSC, have revealed significant destabilization of LcrV after N-terminal deletions [39]. This enlightens the fact that the ionized state of N-terminal amino acids in LcrV protein at lower pH (below 6) and higher temperature leads to destabilization. If the thermo-labile N-terminal region of this plague vaccine antigen is protected using a specific polysaccharide through covalent/non-covalent binding, stability can be achieved. Hence, dextran, which belongs to the polysaccharide group that ionizes in the same pH (pH 7), was chosen. By including dextran and NaCl, particle agglomeration, antigen secondary, and tertiary structural changes were prevented following continuous exposures to 30 °C. The optimum dextran-protein ratio was 1:1, as this enhanced the coupling chemistry to be more efficient, i.e., along with NaCl. The dextran has an increasing number of functional groups per molecule of protein, leading to more bonds between each protein molecule and the surface. As described by Jacob et al., after the molecular crowding by dextran layers with NaCl and LcrV (-NH2 with Cl-), the formation of stable structures must have been attained by the reaction of the carbohydrate hydroxyl groups of dextran with a sodium ion of NaCl in aqueous solution [40]. Dextran and Ficoll 70 have been established as ideal models for principal crowding components in living cells where protein folding occurs [41,42]. This is because their interaction with proteins in the cells can be determined through excluded-volume models. The free energy of unfolding of FK506 binding proteins has been proved to increase with the addition of these two crowding agents [43,44]. Thus, the coupling of LcrV and dextran by the more efficient and site-directed amide-based chemistry has provided more bonds to the surface and a more homogeneous distribution as expressed in UV and FTIR data.

Since the primary objective for a stable formulation of a bacterial protein-based vaccine is to maintain structural integrity and its biological activity, the selection and development of appropriate techniques that detect physical changes directly or indirectly related to bacterial protein conformation have been chosen. The ensilication method suggested by Chen and crew [45] gave a better solution to avoid the cold-chain problem. Their study confirmed the conformation of the secondary and tertiary structure of the Tetanus Toxoid antigen in the formulation by various analytical techniques viz DLS, X-ray diffraction (XRD), CD, FT-IR, etc., and by biological assays (SDS-PAGE, ELISA). Comparing other studies that failed to prove the stability of antigens in the formulation, this research gave knowledge for working in this field. With this reference, the techniques described herein appear to have the capability of monitoring physical events reflecting changes in structural conformation, thus facilitating stability prediction. Even though the changes observed with each of the spectroscopic methods do not directly report on biological activity (unlike invivo methods), the combined results from different techniques seem to provide complementary information from different molecular levels and thus generate an informative overall stability profile of the LcrV formulation at room temperature (30 ± 2 °C). The invivo results also showed promising immunological activity of the LcrV formulation stored at 30 °C for three weeks similar to the previously reported titer values [3,46,47] emphasizing the fact that both structural stability as well as the biological activity of the proposed protein has been retained. Generally, sub-unit vaccine antigens are poorly immunogenic and require an adjuvant to boost the immune response [48]. Interestingly, the formulated protein showed the same or even improved immune response without an adjuvant, hinting at the fact that the biocompatible polysaccharide dextran (70,000 Da) can perform a dual role of stability and adjuvanticity.

Our perception of the mechanism of protein stabilization by polymer and co-solvent interaction was initiated with reference to work performedby Timasheff and his peer workers [49,50]. According to them, the stabilizing effect of polymers with co-solvents towards protein is achieved by four inter-related mechanisms, namely: cohesive force/surface tension mechanism, excluded volume effect, unfavorable interaction with peptide bonds, and preferential exclusion from the protein surface. In the case of liquid formulations, polymers provide stabilization to protein by both proteins-specific and non-specific mechanisms. The polar and hydrophilic polymers such asdextran, poly ethylene glycol (PEG), etc., stabilize proteins by various mechanisms among which, the molecular crowding effect plays a vital role. The same mechanism has worked out in the stabilization of our protein of interest (LcrV) by dextran and NaCl at room temperature (30 ± 2 °C) (refer to graphical abstract). The 2D structure of dextran has been downloaded from PubChem [51], and the 3D structure of LcrV has been retrieved from the SWISS-MODEL repository [52] in the graphical abstract.

Dextran has been reported to stabilize proteins by raising the glass transition temperature (Tg) of the protein formulation [53]. However, in our case, it was unable to effectively hydrogen bond to the LcrV due to steric interference (as discussed earlier). To overcome this, NaCl has been used concurrently. In agreement with another research [54], ionic bonding between salt and protein surfaces may be reduced due to the partitioning of some fraction of the salt molecules with the polymer. Hence, a saturation of salt molecules occurs to interact with and stabilize the protein, which is when the polymer takes the stabilizing role thereafter. So, the effective stability of LcrV cannot be achieved with NaCl alone or dextran alone, which has been articulated in the analytical techniques performed in this work. These are the reasons for us to obtain stability of LcrV only when adding dextran as a major stabilizer and NaCl as a co-stabilizer. The proposed excipients have not only enhanced the structural stability but have also enhanced the biological activity (immune response) of LcrV even after exposure to 30 ± 2 °C for three weeks.

## 4. Conclusions

In this work, the LcrV protein, a highly thermally unstable vaccine candidate for thr plague, has been stabilized using biocompatible polysaccharide dextran at room temperature (30 ± 2 °C) for around three weeks. The stability of LcrV based on protein–polymer interaction was analyzed using various analytical tools such asUV–Visible spectrophotometer, fluorescence spectrophotometer, Fourier transform infrared spectrometer, particle size distribution, gel electrophoresis, transmission electron microscopy, differential scanning calorimetry, and circular dichroism. Based on the UV analysis, a peak at 280 nm corresponding to the aromatic amino acid was obtained in the formulation with hypochromicity observed after a few hours, without change in adsorption maxima, representing the masking or molecular crowding of LcrV by the stabilizers, to provide stability. Based on the fluorescence spec, the emission maxima wavelengths (λmax) and the ratio of the intensity at 345 nm gave an indication of the compactness of the protein in formulations. The FTIR analysis gave a clear background on the possible molecular interactions between LcrV and the stabilizers in enhancing the protein’s secondary structure stability. Next, from the particle size analysis, it was clear that the formulation showed a 100% size distribution (by intensity) with mono dispersion, proving the molecular crowding of the protein by the polymer and the co-excipient, thereby enhancing stability. Based on gel electrophoresis, the formulation showed stability for up to three weeks, beyond which degradation was observed. TEM analysis was performed to ensure the presence of protein in the formulation, as masking of the protein by the stabilizers was observed in all the analyses. The TEM proved the presence of protein (LcrV) in the formulation. Finally, DSC (for melting and enthalpy change) and circular dichroism (for the secondary structure compactness) were performed in the third week to determine the thermal stability of the protein in the formulation and the results ensured the protein stability. Apart from the structural stability, the biological activity of the formulated LcrV stored at room temperature (30 ± 2 °C) for three weeks, in comparison to native LcrV (stored at 4–8 °C) was also performed invivoby ELISA. A significant difference in humoral immunity (IgG endpoint titer) of the formulation (PD1) was observed when compared to the adjuvanted LcrV. It was remarkable to note that the biocompatible polymer dextran had given nearly the same or improved humoral immune response (IgG titer) when compared to alum adjuvanted LcrV. This gives considerable hope to the adjuvant effect of the dextran [55], which has to be explored further.

## Figures and Tables

**Figure 1 vaccines-11-00027-f001:**
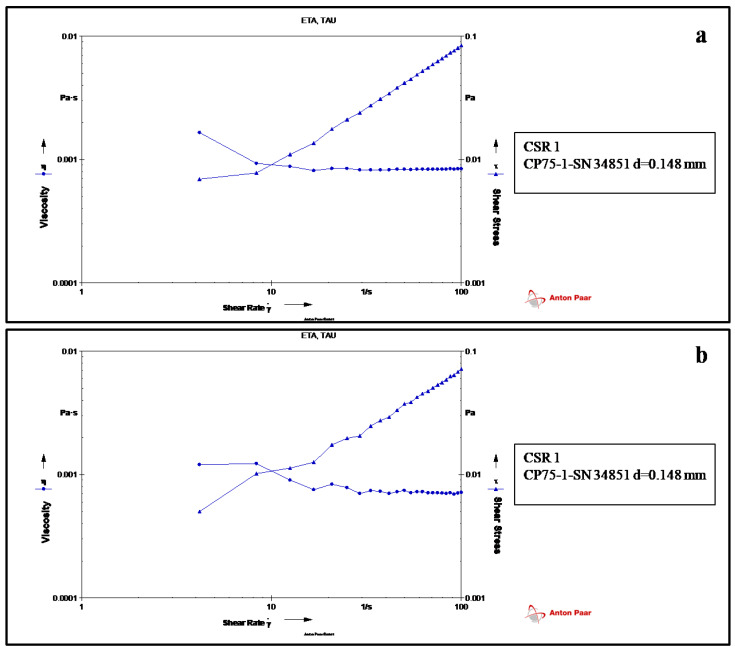
Rheology measurement displaying viscosity and stress of (**a**) dextran and (**b**) formulation (PD1).

**Figure 2 vaccines-11-00027-f002:**
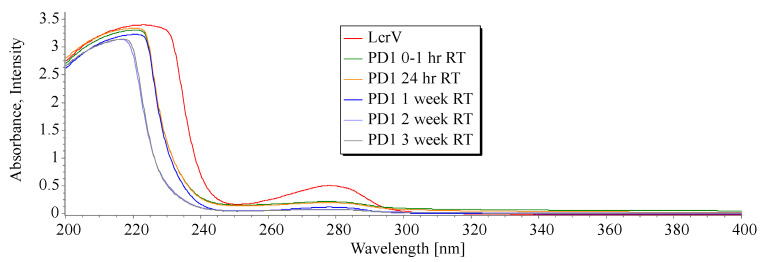
UV–Visible spectra of LcrV and PD1 formulation (at different storage intervals).

**Figure 3 vaccines-11-00027-f003:**
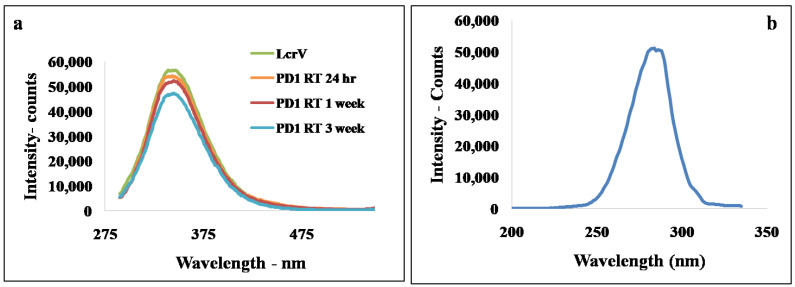
Fluorescence spectroscopic images showing (**a**) emission spectra of samples excited at 278 nm (**b**) excitation spectra with emission monochromator set at 345 nm.

**Figure 4 vaccines-11-00027-f004:**
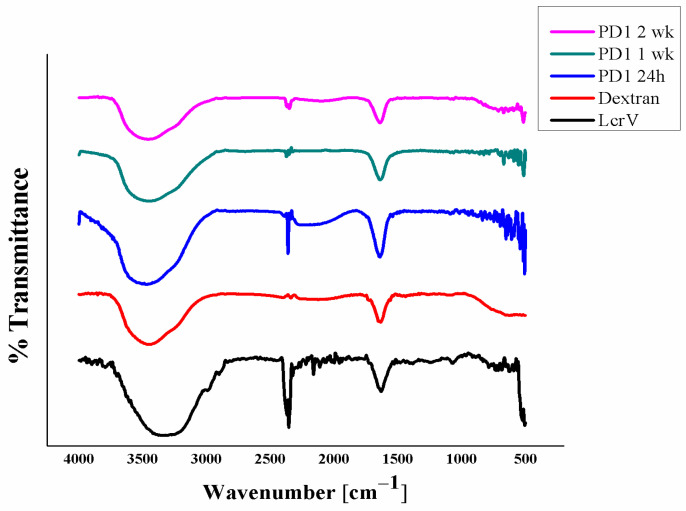
FT-IR spectra of pure protein, polymer, and protein formulation PD1 (after different storage intervals).

**Figure 5 vaccines-11-00027-f005:**
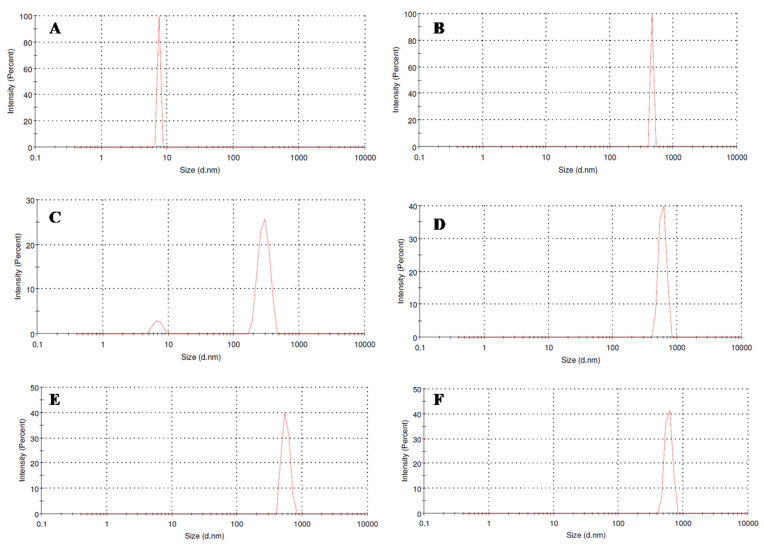
Particle size analysis of (**A**) LcrV, (**B**) dextran, (**C**) formulation PD1 within an hour, (**D**) PD1 after an hour at 30 °C (room temperature), (**E**) PD1 after 1 week of storage at 30 °C, and (**F**) PD1 after 3 weeks of storage at 30 °C.

**Figure 6 vaccines-11-00027-f006:**
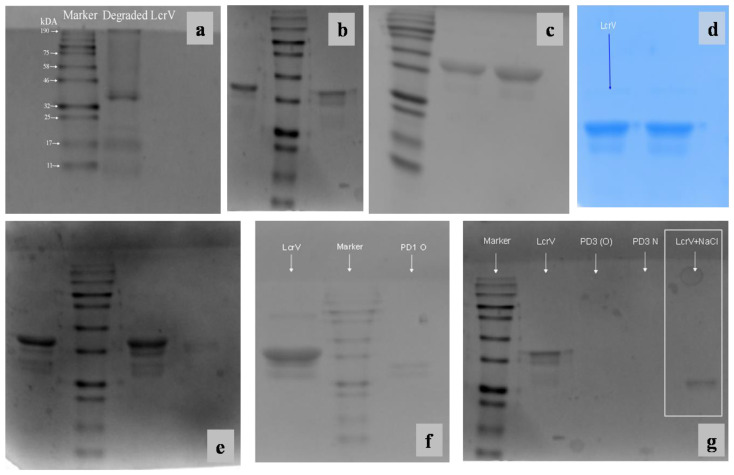
SDS-PAGE analysis of LcrV and PD1 formulations stored at room temperature (30 ± 2 °C) (**a**) Lane 1: Pre-stained protein marker (11–190 kDa); Lane 2: Degraded LcrV after storage at 30 °C (**b**) Lane 1: PD1 formulation at 30 °C for one hour; Lane 2: Protein standard; Lane 3: LcrV (from 4–8 °C) showing band at 37 kDa(**c**) Lane 1: Protein standard; Lane 2: LcrV; Lane 3: PD1 formulation at 30 °C for 24 h (**d**) Lane 1: LcrV; Lane 2: PD1 formulation at 30 °C for 1 week (**e**) Lane 1: LcrV; Lane 2: Protein standard; Lane 3: PD1 formulation at 30 °C for 3 weeks; Lane 4: PD1 formulation at 30 °C after 3 weeks (23 days) (**f**) Lane 1: LcrV; Lane 2: protein standard; Lane 3: Degradation of LcrV in formulation after 3 weeks (band below 32 kDa). (**g**) Lane 1: Protein standard; Lane 2: LcrV; Lane 5: LcrV formulation with NaCl alone showing degradation after 8 days (band at ~25 kDa).

**Figure 7 vaccines-11-00027-f007:**
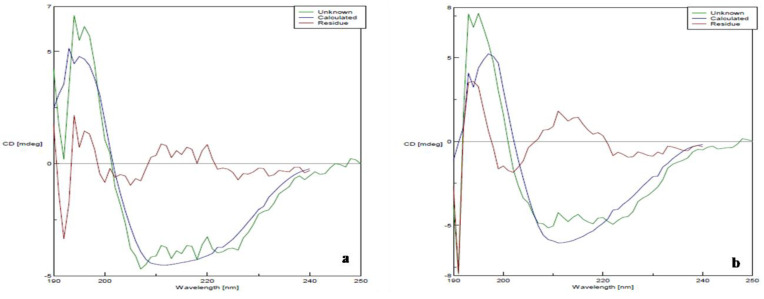
CD spectra of (**a**) LcrV and (**b**) formulation PD1 after 3 weeks in the “far-UV” spectral region.

**Figure 8 vaccines-11-00027-f008:**
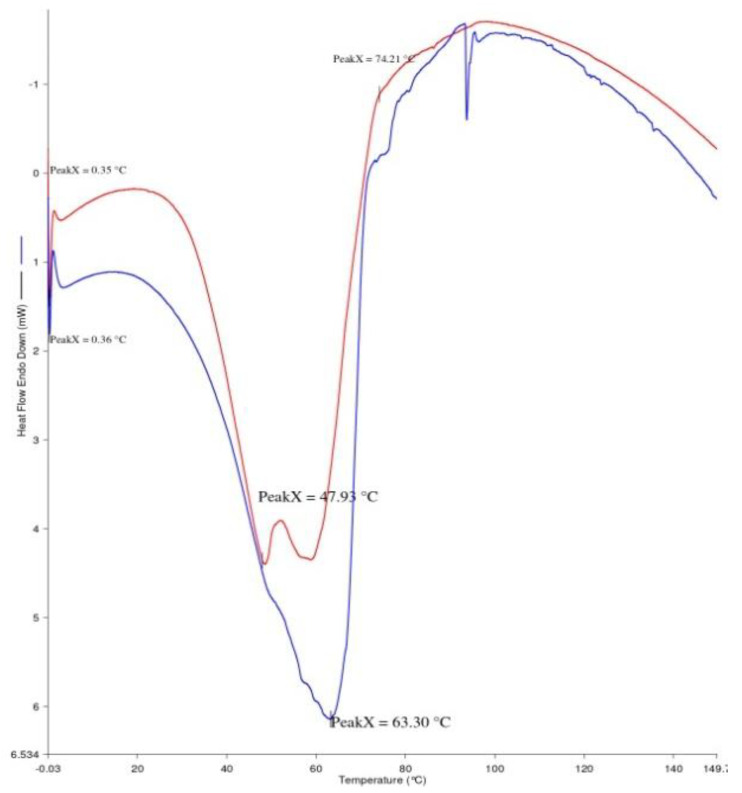
DSC thermogram of LcrV (red) and PD1 formulation (blue) after 3 weeks.

**Figure 9 vaccines-11-00027-f009:**
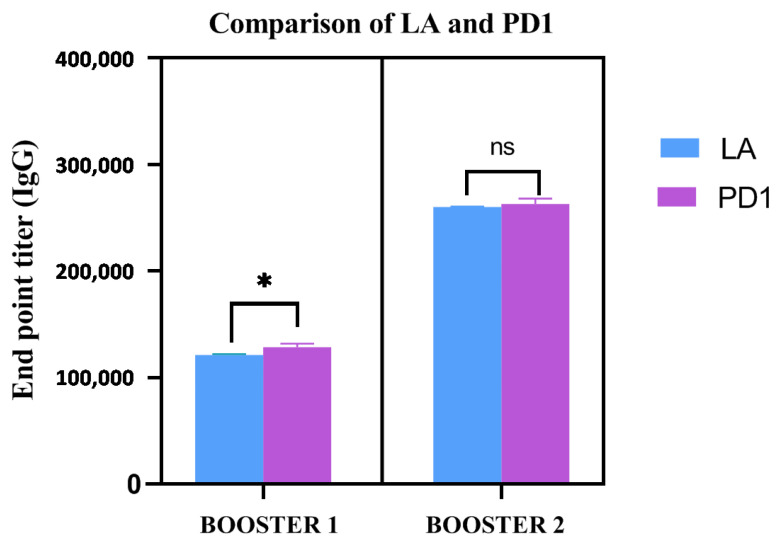
Endpoint titers of IgG antibody estimated by ELISA from serum collected after first and second boosters from vaccinated animal groups, i.e., LA (alum adjuvanted LcrV) and PD1 (formulation stored at 30 °C for three weeks). *n*=5 for each group. * *p*-value = 0.047, ns represents no significance.

## Data Availability

All data generated or analyzed during this study are included in this published article and its Appendix A.

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
