# Peer review of "Determination of Conformational and Functional Stability of Potential Plague Vaccine Candidate in Formulation"

_vaccines, 2022, doi:10.3390/vaccines11010027_

Round 1
Reviewer 1 Report
Review of MS# 2044844 for Vaccines.
DETERMINATION OF CONFORMATIONAL AND FUNCTIONAL STABILITY OF POTENTIAL PLAGUE VACCINE CANDIDATE IN FORMULATION
Krubha Athirathinam, Selvasudha Nandakumar, Shailendra Kumar Verma and Ruckmani Kandasamy
This paper uses a number of biophysical and immunological techniques to study the thermal stability of a plague vaccine candidate when formulated in a mixture of NaCl and dextran as stabilising agents over extended time at 30±2oC. It is found that this formulation is able to maintain antigenic conformation of that vaccine. In addition, the formulated vaccine shows similar antigenicity to the native antigen adjuvanted in Alum.
The experiments are well described, use appropriate methodology and for the most part the conclusions drawn are supported by the data provided.
The paper is generally well written, although does require some editing for correct use of English language and clarity of understanding.
The subject is of interest to the readership of Vaccines and in my opinion the paper should be suitable for publication, but only after significant modification/clarification.
Specifically, the following points should be addressed:
1. The authors need to be careful and clear on the use of the terms Therapeutic and Prophylactic. In definition Therapeutic means a substance used to treat a disease, Prophylactic refers a substance used to prevent a disease. There are a number of places in the text where these terms are incorrectly used. This needs to be corrected.
2. Lines 55 – 60 need to be made clearer. The authors talk about the WHO being concerned about the potential for use of plague as a bioweapon. This is true. However, the majority of cases occurring are the result of natural infections, there have been very few, if any actual reported use as a weapon.
3. Throughout the manuscript the term “room temperature” is used. This is vague and should be more specifically defined as a constant 30±2oC. Room temperature is unlikely to be constant and will often be perceived as different by people in different geographic locations.
4. The images of SDS-PAGE in Fig. 6 are of poor quality, making it difficult to decide if the conclusions drawn are warranted by the data. Better quality images are needed.
1. The TEM images in Supplemental Fig. 5 are also of poor quality. It is very hard to identify particles. Better quality data should be provided the authors should determine the average particle size (this should be possible in most standard TEM image analysis software) and compare to the particle sizes measured by DLS.
2. The functional data shown in Fig. 9 is convincing that the formulated construct is functionally as good as the adjuvanted native antigen. However, the authors also draw the conclusion that this indicates that the formulation has properties of an adjuvant as well as stabilising the antigenic structure. I am not convinced by this argument. Could it not be the case that because the formulation of the antigen maintains the correct antigenic structure that the addition of an adjuvant is no longer needed for a good immune response? What would happen if the formulated antigen was the adjuvanted. Would this result in an even stronger immune response?
Reviewer 2 Report
In this study, the authors characterized the recombinant LcrV protein of Yersinia pestis for its stability and efficiency. The authors showed full characterization for the peptide, however, animal experiments are not convincing.
Major points
1- Figure 9: the authors should challenge the animal with microbes or antigens and then check the neutralization efficiency
2- Kinetics of IgG should be shown at different time points
3- The authors should discuss the difference and new findings of their study and another study PMID: 33353123
4- Supple Fig 5 TEM are very poor, I do not rhink they could be published.
5- Language editing is required.
Round 2
Reviewer 1 Report
I thank the authors for their responses to my comments and consider that the revised manuscript is now acceptable for publication.
Reviewer 2 Report
No further comments